# Associations between Family Functioning and Symptoms of Attention-Deficit Hyperactivity Disorder (ADHD): A Cross-Sectional Study

**DOI:** 10.3390/healthcare10081502

**Published:** 2022-08-09

**Authors:** Yanee Choksomngam, Wichuda Jiraporncharoen, Kanokporn Pinyopornpanish, Assawin Narkpongphun, Krongporn Ongprasert, Chaisiri Angkurawaranon

**Affiliations:** 1Department of Family Medicine, Faculty of Medicine, Chiang Mai University, Chiang Mai 50200, Thailand; 2Department of Psychiatry, Faculty of Medicine, Chiang Mai University, Chiang Mai 50200, Thailand; 3Department of Community Medicine, Faculty of Medicine, Chiang Mai University, Chiang Mai 50200, Thailand; 4Global Health and Chronic Conditions Research Group, Chiang Mai University, Chiang Mai 50200, Thailand

**Keywords:** attention-deficit hyperactivity disorder, ADHD, symptom, family functioning

## Abstract

Poor family functioning is linked with poor child ADHD symptoms. However, there are many dimensions of family functioning. Thus, this study aims to find the association between each dimension of family functioning and controlled ADHD symptoms in an Asian culture. This cross-sectional study involved caregivers of 113 Thai children with ADHD ages 4–16 years old who visited the Outpatient Department at Maharaj Nakorn Chiang Mai Hospital between October 2017 and March 2018. The caregivers completed the Chulalongkorn Family Inventory and the SNAP-IV Thai version. Logistic regression was used to examine each dimension of family functioning adjusting for potential confounders. In univariate analyses, six of the seven dimensions of family functioning were associated with controlled ADHD symptoms. In multivariate analyses, findings revealed that good family roles (aOR 7.48, 95% CI = 1.56 to 35.85, *p* = 0.01) and behavior control (aOR 2.56, 95% CI = 1.08 to 6.03, *p* = 0.03) were associated with controlled ADHD symptoms. In children with ADHD with poor symptom control, the assessment of family functioning could be helpful. Developing a more specific intervention for caregivers that promotes good family roles and behavioral control may be beneficial.

## 1. Introduction

ADHD is one of the most common neurodevelopmental disorders of childhood. In 2016, the National Survey of Children’s Health (NSCH) estimated the number of children and adolescents in the United States ever diagnosed with ADHD to be 9.4% (approximately 6.1 million children) [1]. In 2012, the national prevalence of attention-deficit/hyperactivity disorder (ADHD) among Thai children was 8.1% (approximately 1 million children) [2]. ADHD is characterized by a persistent and pervasive pattern of inattention and/or hyperactivity–impulsivity [3]. The symptoms, particularly if not adequately treated, could have significant negative impacts on the patients’ school, family, and social functioning [4,5]. Conversely, if parents fail to take care of their children in an appropriate way, symptoms of ADHD may worsen [6]. Thus, family functioning is an important aspect of everyday life for children with ADHD and their families as they play a key role in influencing the behavior of children with ADHD [7].

The McMaster Model of Family Functioning was proposed by Epstein in 1987. He regards the process of the family system as the core and assumes that the basic function of the family is to provide appropriate environmental conditions for family members to develop in physical, psychological, social, and other aspects [8]. This model categorized the family functioning into six dimensions which were found to be clinically informant when dealing with families. The six dimensions are: (1) Problem solving—refers to the family’s ability to resolve problems at a level that maintains effective family functioning, (2) Communication—refers to the ability to communicate clearly and directly with other family members, (3) Family roles—refers to how families allocate responsibilities, clarity of household tasks and, provision of resources, (4) Affective responsiveness—refers to the degree of the emotional response of the family members to stimuli, (5) Affective involvement—refers to the degree of concern and attention of family members to each others’ activities and interests, and (6) Behavior control—refers to the ability of family members to set and abide by rules and standards of behavior.

Previous research has found that more negative family functioning is linked with more child ADHD symptoms [6]. However, little is known about the association between each dimension of family functioning and controlled ADHD symptoms. Knowing which specific dimension contributes to controlled ADHD symptoms can help guide specific interventions for patients and their families. Moreover, evidence of the effects of family functioning on ADHD may not apply to different settings. There is some evidence that the cultural environment may affect a child’s behavior and may influence perceptions with regards to the subjective assessment of ADHD symptoms [9,10,11]. One study from Korea suggested that the prevalence of ADHD in Asian countries may be lower than those reported in the US since children in Asian cultures are taught to act and speak quietly in public [10].

Thus, this study aims to find the association between each dimension of family functioning and controlled ADHD symptoms in Thailand, which can lead to better ways of taking care of children with ADHD and their families, especially in the Asian context.

## 2. Materials and Methods

This cross-sectional study involved caregivers of 113 ADHD children ages 4–16 years old who visited the Outpatient Department of Child and Adolescent Psychiatry, Maharaj Nakorn Chiang Mai Hospital, Thailand (a university hospital in an urban area), between October 2017 and March 2018. Self-administered questionnaires were completed by caregivers, including on demographic characteristics (caregiver’s sex, caregiver’s age, caregiver’s education, caregiver’s stress, child’s sex, child’s age, medication, duration of treatment in this hospital, family formation, and number of siblings).

The caregiver’s stress was assessed using the Suan Prung Stress Test (SPST). The SPST is the standard questionnaire recommended by the Department of Mental Health, Ministry of Public Health. Cronbach’s alpha reliability coefficient was more than 0.7. The SPST consists of 20 questions, each with a score of 1 to 5 (higher scores reflect severe stress). The range of overall scores was divided into two levels (normal stress and severe stress). The cut-off point for severe stress was more than 43 [12].

(1)Family functioning was assessed using the Chulalongkorn Family Inventory (CFI), which assessed the six dimensions as proposed by the McMaster Model of Family Functioning [13]. The CFI is a self-report questionnaire in the Thai language that has been widely used in Thailand because of its good reliability (alpha 0.88). The CFI consists of 36 questions measuring family function in 6 dimensions according to the McMaster model, i.e., problem-solving (6 questions), communication (5 questions), family roles (3 questions), affective responsiveness (5 questions), affective involvement (5 questions), and behavioral control (4 questions). In addition, the CFI included one additional dimension, the general overall functioning dimension (eight questions), as an overall surrogate measure of family functioning. For each question, scores range from 1 to 4, with higher scores reflecting healthier family functioning. The scores for the individual questions for each dimension were then combined to give an overall score for each dimension. According to the CFI, the scores for each dimension can be categorized into four levels (poor, fair, average, and good). For analyses, due to the small sample sizes for those scoring at the first and second levels, these two levels were combined and categorized as poor and the remaining were categorized as fair (third level) or good (fourth level).(2)Child inattention and hyperactivity/impulsivity symptoms were assessed using the Thai version of SNAP-IV [14]. Cronbach’s alpha reliability coefficients were 0.93–0.96. This test consists of 26 items corresponding to criterion A of the DSM-IV for ADHD and the symptoms of the oppositional defiant disorder (ODD). It is separated into three subscales, i.e., inattention (items 1–9), hyperactivity/impulsivity (items 10–18), and opposition/defiance (items 19–26). For each item, scores range from 0 to 3, with higher scores reflecting more severe symptoms. The sum scores of each subscale were then dichotomized to indicate clinical significance or its absence of the symptoms on any of the three subscales. For this study, the condition of controlled ADHD symptoms was that treatment decreased symptoms on both the inattention and hyperactivity/impulsivity subscales (<16 points on the inattention subscale and <14 points on the hyperactivity/impulsivity subscale from the Thai version of SNAP-IV) as these are commonly used in clinical practice and the literature [15,16].

### 2.1. Sample Size Calculation

Based on a published finding that the prevalence of controlled ADHD symptoms was approximately 24% [17] and assuming an alpha error of 5% and error (d) of 10%, it was estimated that at least 71 records were needed. However, we reviewed all records available during the study period.

### 2.2. Statistical Analyses

We used STATA version 16 (StataCorp. Statistical Software: Release 16. College Station, TX, USA: StataCorp LLC.) for data analyses. Characteristics of children with ADHD, their families, and family functioning were reported as numbers and percentages for the categorical data and as means and standard deviations or medians and interquartile ranges as appropriate for the continuous data. For the univariate analyses, we used Fisher’s exact test, chi-square, and *t*-test. Fisher’s exact test and chi-square were used to explore independency in the categorical variables between the two groups. The *t*-test was used to compare means between groups and explore factors associated with controlled ADHD symptoms. For multivariate analyses, logistic regression was used to identify the associations between family functioning and ADHD symptoms adjusting for predefined potential confounders as identified in the literature [4,17,18,19], consisting of caregiver’s sex, caregiver’s age, caregiver’s education, caregiver’s stress, child’s sex, child’s age, medication, duration of treatment in this hospital, family formation, and number of siblings. Results were presented as adjusted odd ratio, 95% confident interval, and *p*-value, and *p*-value of less than 0.05 is considered to be statistically significant. A sensitivity analysis was also conducted to explore the associations between family functioning and symptom control for each of the two subscales of ADHD (inattention and hyperactivity/impulsivity) separately.

## 3. Results

The mean age of the caregivers was 43 years, most were female (81%), 49% held a bachelor’s degree, and 26% reported severe stress (Table 1). The mean age of the children was 10 years, and most of them were male (86%), had one sibling (64%), and came from nuclear families (53%). The median duration of treatment was 26 months (IQR 10–42). The most common medication prescribed was methylphenidate (immediate-release), followed by antipsychotics, methylphenidate (extended-release), antidepressants, and atomoxetine (Table 1).

### Family Functioning and Controlled ADHD Symptoms

In Table 2, the assessment of family functioning showed that many caregivers reported good family functioning for problem-solving (43%), family roles (70%), affective responsiveness (47%), and general overall functioning (57%). However, most of the caregivers also reported poor family functioning for behavior control (43%). Most individual dimensions of family functioning, except for behavioral control, were significantly associated with controlled ADHD symptoms. After adjusted for potential confounders, only two dimensions remained significant: Good family roles (aOR 7.48, 95% CI = 1.56 to 35.85, *p* = 0.01) and behavior control (aOR 2.56, 95% CI = 1.08 to 6.03, *p* = 0.03) were individually associated with controlled ADHD symptoms (Table 3).

The associations between family functioning and controlled ADHD symptoms for each subscale (inattention and hyperactivity/impulsivity subscales) are summarized in Table 4 and Table 5, respectively. The findings revealed that good family roles (aOR 11.29, 95% CI = 2.33 to 54.78, *p* = 0.003) were associated with controlled ADHD symptoms on the inattention subscale. However, there were no associations between family functioning and controlled ADHD symptoms on the hyperactivity/ impulsivity subscale.

## 4. Discussion

Among 113 ADHD children, 29% of them have controlled ADHD symptoms. While most dimensions of family functioning were associated with controlled ADHD symptoms, the study revealed that family roles and behavior control dimensions have been a key factor in the better control of ADHD symptoms.

This study revealed that 29% of children with ADHD had controlled ADHD symptoms. This finding was similar to the result from research in Eastern Thailand which reported that 24% of children with ADHD had controlled behavior [17]. Moreover, the behavioral problems of children with ADHD were less severe in families with good family functioning [17]. From the multivariate analysis, it was found that good family roles and behavior control were associated with controlled ADHD symptoms. These two dimensions of the McMaster Family Functioning Model provide a structured environment, including consistent house rules, expectations, and consequences clearly understood by all family members. This structured environment through good family roles and behavior control may provide benefits in maintaining self-control for children with ADHD who are characterized by difficulty in self-regulation, lack of persistence, and disorganization [20,21].

Specifically, good family roles refer to the provision of resources, supporting personal development, and maintaining and managing the family systems [8]. This dimension was associated with controlled ADHD symptoms because it was likely to be the opposite of the problematic parenting styles that often occur in families with ADHD children, such as being less attentive to the overall needs of their children and poorer monitoring of symptoms [22]. Good behavior control refers to the way the family expresses and maintains standards for the behavior of its member [8]. This dimension was associated with controlled ADHD symptoms because it is also likely decreases problematic parenting styles such as less consistent parenting strategies, less consistent discipline, and laxness in parenting [22].

When considering the associations between family functioning and controlled ADHD symptoms for each subscale, a positive association between good family roles and controlled ADHD symptoms inattention subscale was observed, but no association was found for the ADHD hyperactivity/impulsivity subscale. Good family roles are likely more specific to the core symptoms of ADHD inattention type, which are characterized by difficulties sustaining attention in tasks, following through on instructions, finishing chores, and organizing tasks [20]. On the contrary, the literature suggests that hyperactivity/impulsivity symptoms often diminish with age [23], and thus, family functioning may not play a key role in this specific subscale of symptoms.

Overall, our findings support that family functioning is an important factor for achieving the control of ADHD symptoms. They provide further insights that family roles and behavior control are the important dimension of family functioning, which supports the control of ADHD symptoms in an Asian context. These findings are consistent and help support the theory that well-established behavioral treatment for children with ADHD and training for parents in behavior management should be applied and is appropriate to Asian settings [19,24]. For effective treatment, health personnel should support and empower the families of children with ADHD by training care providers to create good family roles. If the family applies the training consistently, this increases the chance of repeated practice with performance feedback over time. Repeated practice subsequently affects behavior better than modifying behavioral contingencies in a specific setting [19].

A limitation of this study is that ADHD symptoms were reported from only one source, namely one caregiver of children with ADHD, which may cause bias. A previous study showed that mothers tended to report more symptoms than fathers [25]. The questionnaires were self-administered, so there may be some biases if the participants did not understand the meaning of each item or did not answer truthfully. However, self-administered questionnaires from a single caregiver are routine for assessing ADHD symptoms and are used for clinical management in our setting. Research assistants who were not part of the clinical team were also available to clarify any parts of the questionnaires that the caregiver may have had questions about. We used a Thai-specific tool for the assessment of family function. This has been validated, but it may make comparisons of findings using other tools to assess family function difficult. In addition, this study is limited by study design as causality cannot be inferred in a cross-sectional study. Therefore, future longitudinal research may provide better insights. Finally, the results may not apply to all settings in Thailand because the study was conducted in a single referral center with a small sample size. Larger multicenter validation studies would be useful. A strength of this study is the classification of family functioning into dimensions according to the McMaster Model, which helps us to understand more about the association between each dimension of family functioning and controlled ADHD symptoms. Moreover, this understanding may lead to developing a more specific intervention for children with ADHD and their families in Asian culture.

## 5. Conclusions

Family functioning was associated with controlled ADHD symptoms. In children with ADHD with poor symptom control, assessment of family functioning could be helpful. Developing a more specific intervention for caregivers that promotes good family roles and behavioral control may be beneficial in the Thai setting.

## Figures and Tables

**Table 1 healthcare-10-01502-t001:** Demographic Characteristics of Children with ADHD, Caregivers, and Their Families.

Characteristics	Total (N = 113)	ADHD Symptoms		
		Uncontrolled (N = 80)	Controlled (N = 33)	*p*-Value
Caregiver’s sex (*n*, column %) Male Female	22 (19.47) 91 (80.53)	13 (16.25) 67 (83.75)	9 (27.27) 24 (72.73)	0.20
Mean caregiver’s age (SD)	43.08 (8.63)	42.18 (8.73)	45.27 (8.08)	0.08
Caregivers’ education (*n*, column %) Below bachelor’s degree Bachelor’s degree Above bachelor’s degree	37 (32.74) 55 (48.67) 21 (18.58)	29 (36.25) 40 (50.00) 11 (13.75)	8 (24.24) 15 (45.45) 10 (30.30)	0.12
Caregiver’s stress (*n*, column %) Normal stress Severe stress	84 (74.34) 29 (25.66)	52 (65.00) 28 (35.00)	32 (96.97) 1 (3.03)	<0.01
Child’s sex (*n*, column %) Male Female	97 (85.84) 16 (14.16)	65 (81.25) 15 (18.75)	32 (96.97) 1 (3.03)	0.04
Mean child’s age (SD)	9.67 (3.0)	9.57 (3.05)	9.91 (2.91)	0.59
Current Medication (*n*, column %) Methylphenidate (immediate-release) Methylphenidate (extended-release) Atomoxetine Antidepressant Antipsychotic	59 (52.21) 22 (19.47) 4 (3.54) 15 (13.27) 26 (23.01)	44 (55.00) 14 (17.50) 2 (1.60) 12 (9.60) 19 (15.2)	15 (45.45) 8 (2.64) 2 (0.66) 3 (0.99) 7 (2.31)	0.41 0.44 0.58 0.55 1.00
Mean Duration of treatment in months (SD)	31.50 (29.28)	31.46 (31.13)	31.58 (24.68)	0.98
Median Duration of treatment (month) (IQR)	26 (10–42)	25.5 (6.5–43.5)	28 (15–42)	0.51
Family formation (*n*, column %) Nuclear family Extended family Single dad/ mom	58 (52.73) 40 (36.36) 12 (10.91)	37 (47.44) 31 (39.74) 10 (12.82)	21 (65.63) 9 (28.13) 2 (6.25)	0.24
No. of Siblings (*n*, column %) 0 1 2	40 (35.40) 72 (63.72) 1 (0.88)	30 (37.5) 50 (62.50) 0 (0.00)	10 (30.30) 22 (66.67) 1 (3.03)	0.29

**Table 2 healthcare-10-01502-t002:** Demographic Characteristics of Family Functioning.

Characteristics	Total (N = 113)	ADHD Symptoms		
		Uncontrolled (N = 80)	Controlled (N = 33)	*p*-Value
Problem-solving Good Fair Poor	49 (43.36) 33 (29.20) 31 (27.43)	28 (35.00) 23 (28.75) 29 (36.25)	21 (63.64) 10 (30.30) 2 (6.06)	<0.01
Communication Good Fair Poor	46 (40.71) 52 (46.02) 15 (13.27)	27 (33.75) 40 (50.00) 13 (16.25)	19 (57.58) 12 (36.36) 2 (6.06)	0.05
Family roles Good Fair Poor	79 (69.91) 22 (19.47) 12 (10.62)	48 (60.00) 21 (26.25) 11 (13.75)	31 (93.94) 1 (3.03) 1 (3.03)	<0.01
Affective responsiveness Good Fair Poor	53 (46.90) 42 (37.17) 18 (15.93)	31 (38.75) 32 (40.00) 17 (21.25)	22 (66.67) 10 (30.30) 1 (3.03)	0.01
Affective involvement Good Fair Poor	42 (37.17) 50 (44.25) 21 (18.58)	22 (27.50) 39 (48.75) 19 (23.75)	20 (60.61) 11 (33.33) 2 (6.06)	<0.01
Behavioral control Good Fair Poor	17 (15.04) 47 (41.59) 49 (43.36)	10 (12.50) 31 (38.75) 39 (48.75)	7 (21.21) 16 (48.48) 10 (30.30)	0.16
General overall functioning Good Fair Poor	64 (56.64) 35 (30.97) 14 (12.39)	36 (45.00) 31 (38.75) 13 (16.25)	28 (84.85) 4 (12.12) 1 (3.03)	<0.01

**Table 3 healthcare-10-01502-t003:** Associations between Family Functioning and Controlled ADHD Symptoms (SNAP-IV: both inattention and hyperactivity/impulsivity subscales were controlled).

Family Functioning ^#^	Controlled ADHD Symptoms *		
	aOR	95% CI	*p*-Value
Problem solving	0.69	0.19 to 2.48	0.57
Communication	0.40	0.11 to 1.40	0.15
Family roles	7.48	1.56 to 35.85	0.01
Affective responsiveness	1.38	0.41 to 4.63	0.61
Affective involvement	1.01	0.33 to 3.02	0.99
Behavioral control	2.56	1.08 to 6.03	0.03
General overall functioning	1.76	0.29 to 10.58	0.54

^#^ Each dimension of family functioning has three levels (good, fair, poor) and is treated as a score (0,1,2). * Each dimension of family functioning was individually adjusted for potential confounders (caregiver’s sex, caregiver’s age, caregiver’s education, caregiver’s stress, child’s sex, child’s age, medication, duration of treatment in this hospital, family formation, and number of siblings).

**Table 4 healthcare-10-01502-t004:** Associations between Family Functioning and Controlled ADHD Symptoms (SNAP-IV: inattention subscales).

Family Functioning ^#^	Controlled ADHD Symptoms (Inattention Subscales) *		
	aOR	95% CI	*p*-Value
Problem solving	0.90	0.26 to 3.09	0.87
Communication	0.61	0.17 to 2.25	0.46
Family roles	11.29	2.33 to 54.78	<0.01
Affective responsiveness	1.74	0.50 to 6.06	0.38
Affective involvement	1.20	0.40 to 3.59	0.74
Behavioral control	1.92	0.74 to 5.03	0.18
General overall functioning	0.74	0.13 to 4.07	0.73

^#^ Each dimension of family functioning has three levels (good, fair, poor) and is treated as a score (0,1,2). * Each dimension of family functioning was individually adjusted for potential confounders (caregiver’s sex, caregiver’s age, caregiver’s education, caregiver’s stress, child’s sex, child’s age, medication, duration of treatment in this hospital, family formation, and number of siblings).

**Table 5 healthcare-10-01502-t005:** Associations between Family Functioning and Controlled ADHD Symptoms (SNAP-IV: hyperactivity/impulsivity subscales).

Family Functioning ^#^	Controlled ADHD Symptoms (Hyperactivity/Impulsivity Subscales) *		
	aOR	95% CI	*p*-Value
Problem solving	0.74	0.28 to 1.94	0.54
Communication	0.93	0.36 to 2.39	0.87
Family roles	1.77	0.68 to 4.63	0.24
Affective responsiveness	0.96	0.37 to 2.45	0.92
Affective involvement	1.05	0.43 to 2.52	0.92
Behavioral control	1.45	0.71 to 2.96	0.31
General overall functioning	0.89	0.27 to 2.94	0.84

^#^ Each dimension of family functioning has three levels (good, fair, poor) and is treated as a score (0,1,2). * Each dimension of family functioning was individually adjusted for potential confounders (caregiver’s sex, caregiver’s age, caregiver’s education, caregiver’s stress, child’s sex, child’s age, medication, duration of treatment in this hospital, family formation, and number of siblings).

## Data Availability

The data presented in this study are available on request from the corresponding author. The data are not publicly available due to privacy and ethical restrictions.

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
