# Peer review of "Associations between Family Functioning and Symptoms of Attention-Deficit Hyperactivity Disorder (ADHD): A Cross-Sectional Study"

_healthcare, 2022, doi:10.3390/healthcare10081502_

Round 1

Reviewer 1 Report

This study investigated the relationship between each dimension of family functioning and the severity of ADHD symptoms in Thailand. Family functioning was assessed using the Chulalongkorn Family inventory, which assessed the 6 dimensions, including 1) Problem solving, 2) Communication, 3) Family roles, 4) Affective responsiveness, 5) Affective involvement, 6) Behavior control. ADHD symptoms were assessed using the Thai version of SNAP IV and ‘controlled ADHD symptoms’ were defined as having no clinically significant symptoms in both the inattention and hyperactivity/impulsivity sub-scales. It was found that good family roles and behavior control were individually associated with controlled ADHD symptoms.

1. Participants of the study were mainly outpatients from a single hospital. Clinical samples may be influenced by selection biases and only represent some part of the persons suffering from the ADHD symptoms.

2. The study data were collected from ‘self-administered questionnaires’ completed by participants. It was not sure whether the participants really understood the meaning of each item. Participants might describe their wishes instead of the facts.

3. Questionnaire to assess family function only contains 3 to 6 questions in each dimensions. It could be questioned if the result could present reliable condition of each dimension.

4. The sample size of the study was small. This could limit the statistical power of deciding the association between family function and ‘controlled ADHD symptoms’.

Author Response

Thank you for your comments.  Please see responses in the attached file

Reviewer 2 Report

Major Points:

1.       I had a bit of a struggle with understanding “controlled ADHD” symptoms. At first glance, it seems to be somewhat of a misnomer, since it would suggest fewer/less symptoms and hence less severe ADHD. Later in the paper, it states that these children had no clinically significant symptoms. This raises the question of how can a child have ADHD, and not have clinically significant symptoms? I think in the end, I understand “controlled ADHD” but that should probably be clearer in the introduction of the paper. Is this simply the case that medication decreases symptoms or are the symptoms actually remitted? I have a feeling that other readers will also struggle where I did.

2.       Can you please expand on how the confounders were controlled?

3.       For Tables 2 and 3, can you provide a bit more information about the statistical analyses? This can be done in the relevant results sections or in the statistical analyses section (2.2).

Minor points:

1.       Final sentence of abstract: delete “in the Thai setting”.

2.       Line 127: “univariate”

3.       Line 196: “multivariate”

Author Response

Thank you for your comment, please see our responses and changes in the attached file
